# Influence of an Engineered Notch on the Electromagnetic Radiation Performance of NiTi Shape Memory Alloy

**DOI:** 10.3390/ma17071708

**Published:** 2024-04-08

**Authors:** Anu Anand, Rajeev Kumar, Shatrudhan Pandey, S. M. Mozammil Hasnain, Saurav Goel

**Affiliations:** 1Department of Mechanical Engineering, Birla Institute of Technology, Mesra, Ranchi 835215, India; anuanandynr@gmail.com; 2Department of Production and Industrial Engineering, Birla Institute of Technology, Mesra, Ranchi 835215, India; er.shatrudhanp@gmail.com; 3Faculty of Engineering and Applied Science, Usha Martin University, Ranchi 835103, India; smmh429@gmail.com; 4School of Engineering, London South Bank University, 103 Borough Road, London SE1 0AA, UK; 5Department of Mechanical Engineering, University of Petroleum and Energy Studies, Dehradun 248007, India

**Keywords:** EMR, Nitinol, notch width, plastic zone radius, crack, microstructural analysis

## Abstract

This work explores the influence of a pre-engineered notch on the electromagnetic radiation (EMR) parameters in NiTi shape memory alloy (SMA) during tensile tests. The test data showed that the EMR signal fluctuated between oscillatory and exponential, signifying that the specimen’s viscosity damping coefficient changes during strain hardening. The EMR parameters, maximum EMR amplitude, and average EMR energy release rate remained constant initially but rose sharply with the plastic zone radius with progressive loading. It was postulated that new Frank–Read sources permit dislocation multiplication and increase the number of edge dislocations participating in EMR emissions, leading to a rise in the value of EMR parameters. The study of the correlation between EMR emission parameters and the plastic zone radius before the crack tip is a vital crack growth monitoring tool. An analysis of the interrelationship of the EMR energy release rate at fracture with the elastic strain energy release rate would help develop an innovative approach to assess fracture toughness, a critical parameter for the design and safety of metals. The microstructural analysis of tensile fractures and the interrelation between deformation behaviours concerning the EMR parameters offers a novel and real-time approach to improve the extant understanding of the behaviour of metallic materials.

## 1. Introduction

Metals and alloys under plastic deformation emit emissions in the form of thermal, electric, acoustic, and EMR emissions. The study of real-time EMR signals during loading is vital as it provides a clearer picture regarding the complicated physics involved in crack initiation, nucleation, and propagation.

Notches can have a significant effect on a material’s stress–strain behaviour. A notch can cause stress concentrations, enforcing a plastic state at the notch root. Research on the sensitivity of the notch width and shape is vital to understanding the deformation behaviour of engineering materials [1,2,3].

Misra led pioneering efforts in the 1980s [4,5,6] to report the EMR emission signals in metallic materials during fracture. Several researchers since then have examined this phenomenon [7,8,9,10,11]. During plastic deformation of the uncoated and coated metal sheets, a secondary EMR signal emission phenomenon was discovered [12]. An investigation was carried out to study the influence of the specimen’s axis angle during rolling on the EMR signal emission in real time [13]. For an axis angle of 45°, there were noticeable differences in the magnitude of EMR variations concerning the rolling direction of a brass sheet. No appreciable changes were observed during cutting of the sample in a transverse direction. However, metallic alloys emitted EMR emission signals under compression and plastic deformation [14].

EMR emission parameters during plastic deformation in tension differed significantly from those seen in compression [15]. A smooth correlation was observed between the EMR emission, grain size, and lattice constant [16,17]. Experiments using intermittent EMR in the plastic deformation of Cu-Ni alloys during tension and compression revealed that the compression viscosity coefficient increases during plastic deformation [18]. The plastic zone radius was correlated with the notch–depth ratio in tensile tests conducted on the Cu-Ni alloy [19]. Anand and Kumar’s [20] review of EMR emission and the mathematical models for different metallic materials provides a solid foundation on this topic. In recent years, the EMR emission phenomenon in several metallic materials has been investigated [21,22,23].

The influence of the notch–width ratio on EMR parameters during tensile induced fracture of C35000 brass was recently reported [24]. As a follow-up to this recent report, this investigation was directed at NiTi shape memory alloy (55.7 wt.-% Ni, 44.09 wt.-% Ti).

Being an essential biomedical material, Nitinol has received strong interest from the community in studying its mechanical properties in harsh environments, including investigations on its elastic modulus, strength, and fracture toughness. Nitinol has a lower elastic modulus that matches closely with natural bone material, which makes it an ideal choice for biomedical applications. Nitinol is also suitable for devices experiencing cyclic deformation with high strain recovery of around 10% during deployment. Moreover, NiTi possesses exceptional properties such as superelasticity, shape memory effect, and distinct properties like damping characteristics [25]. Nitinol’s stress–strain behaviour closely resembles human body structures [26]. These applications motivated this research on studying EMR during deformation of NiTi.

As such, EMR is a non-destructive tool for analysing crack initiation and propagation in metallic materials. It is proposed that Nitinol can be used to make Intelligent Reinforced Concrete (IRC). Nitinol is a superelastic material capable of reversing stress-induced martensite transformation, returning it to its original neutral austenite state [25,26]. Nitinol usually resides in austenite and martensite phases. Austenite (B2 phase) is stable at higher temperatures and possesses a body-centred cubic structure. At lower temperatures, the martensite (FCC) structure is more stable. Phase transformation can be due to a shift in temperature or applied stress. These wires can sense cracks and contract to heal macro-sized cracks. The relationship between EMR emissions and associated mechanical parameters at the atomic level is vital [20]. The current research offers novel insights into the tensile fracture behaviour of Nitinol, shedding light on the intricate relationship between crack initiation, fracture behaviour, and EMR emission parameters at varying notch dimensions.

## 2. Materials and Methods

Nitinol sheets procured from Siddhgiri Tubes, Mumbai, were chosen for the investigations. The sheets, with dimensions of 70 mm × 12 mm × 0.50 mm, were cut along the longitudinal axis aligned with the rolling direction using the Wire Electrical Discharge Machine (WEDM) facility at the Jharkhand tool room, Ranchi. According to the certificate provided by the suppliers, Table 1 provides the material composition.

For the EMR emission tests, double semicircle edge notches at the specimen’s centre on each end facilitated the analyses of the emission of EMR signals at tensile fracture.

For studies investigating the correlations between the EMR and mechanical parameters, the notch length “*a*” was set to 0.5, 1, 1.50, 2, 2.50, and 3 mm, respectively. The test specimen had a width (*w*) of 12 mm. Figure 1a,b show the specimen configuration and Nitinol test specimen for the experimental investigations.

Fracture analyses restrain the parameter 2*a*/*w* (notch–width ratio). If *w*2*a* is too minimal compared to the plastic zone size during fracture, the quasi-elastic solution for stress intensity fails. The failure occurs because of the influence of a stress-free boundary on the stress field surrounding the fracture tip. As a result, it is suggested that an ideal 2*a*/*w* range of 0.45 to 0.55 [19] is maintained.

### 2.1. Instrumentation

The tensile fracture testing was performed on an Instron 8801 servo-hydraulic testing machine. The samples were double-notched semicircles with different 2*a*/*w* values. The tests were performed under continuous loading until fracture. The experimental plan with the input used in testing is shown in Table 2. NiTi test specimens of varying notch–width ratios and constant thickness were used for the tensile test. A constant loading rate of 5 mm/min was maintained for the test. The gauge length was taken as 70 mm. Several trial runs were performed to select the input parameters. The repeatability and distribution of EMR signals in metals are essential for various applications, including non-destructive testing (NDT), material characterisation, and quality control. Even modest differences in the provided sample notches led to a significant variance in the EMR amplitude. Also, fracture test data have been found to be distributed widely. Alternatively, due to the EMR emission happening at the atomic level, relying on statistical averaging or the standard deviation approach for analysis is not advisable. Therefore, trials were conducted multiple times within each sample, and the EMR signals revealing the highest consistency were selected for analysis.

The loading rate influences the tensile properties and fracture behaviour. Hence, standard loading rates are required for tensile testing. A higher loading rate of 5 mm/min for the test was selected considering Nitinol’s high tensile strength, superelastic nature, and proper capturing of the EMR signal. A thermal insulation sheet was pasted on both sides of the test sample. Two 12 mm × 10 mm × 0.2 mm copper chips were glued near the notch surface. The insulation paper prevented the antenna’s direct contact with the specimen’s surface. The specimen’s surface and copper chips created a parallel plate capacitor. The electrical wires were soldered on the two copper chips to act as an antenna. The antenna linked the specimen’s signal-emitting dipoles and the digital Tektronix Inc., Beaverton, Oregon, United States, DPO2022B model oscilloscope. The EMR emission signals of the test specimen were recorded using the antenna at fracture. The EMR signals were captured through the experimental setup shown in Figure 2.

The voltage probe of the oscilloscope was attached to the electrical wires from the copper chips. One end of the probe was grounded, and the other was connected to the antenna. The oscilloscope captured the EMR signal sensed by the antenna. The voltage scale of the oscilloscope was set to 20 mV/division and the time scale to 1 s/division. Three-layered 0.5 mm silicon steel is also known as electrical steel due to 3 to 4 wt % per silicon added in iron. Silicon steel sheets widely used for magnetic shielding due to their capacity to absorb magnetic fields were sandwiched between 0.3 mm copper sheets enclosing the test specimen beside the two axial ends, acting as an electromagnetic shield. This minimised the effect of electromagnetic and external noise. After the experimentation, the oscilloscope data were stored on a 32 GB USB drive for further analysis through MATLAB (r2016a) software.

### 2.2. Assessment of Mechanical and EMR Parameters

In fracture mechanics, the stress intensity factor, *K_I_*, estimates the stress intensity around the crack tip or notch under loading. *K_I_* for the double-notched semicircular specimen configuration in the tensile fracture mode is calculated using Equation (1) [24].
(1)KIσπa=1.1220.933+0.1802aw−1.0602aw2+1.7102aw3
where *σ* is the maximum tensile stress applied at the crack tip instability. 

*K_I_* reaches a critical value (*C*) in the mode 1 tensile test such that fracture occurs if *K_I_* = *K_C_*. The *K_C_* values are shown in Table 3.

Under continuous load, a plastic zone forms before the crack develops fully. Edge dislocations created by the crack tip are temporarily pinned in the plastic zone and experience time-dependent bending, leading to accelerated electric line dipoles and the detected EMR emissions [6]. Hence, EMR emissions must be affected based on the formation of the plastic zone before the crack tip. Equation (2) [27] determines the plastic zone radius, *r_y_*.
(2)$$r_{y}=\frac{1}{2\pi }{\left(\frac{K_{I}}{\sigma_{y}}\right)}^{2}$$

The values of the yield stress *σ_y_* of the specimen were found from the experiments to be about 500, 566.7, 600, 700, 733.3, and 800 MPa, respectively. Notches, holes, or other geometrical features can create stress concentrations, lowering the measured yield stress compared to a specimen without such features. Stress concentrations can trigger earlier onset of plastic deformation. The *r_y_* values at different notch–width ratios (2*a*/*w* = 0.08, 0.16, 0.25, 0.33, 0.42, and 0.50) were calculated for each specimen at maximum tensile load at maximum EMR emission (Table 3). As the 2*a*/*w* value increases, the size of the plastic zone formed before the crack tip increases. As an edge notch crack advances toward the centre, it encounters a larger elastic–plastic boundary during progressive deformation in materials with a lower 2*a*/*w* value; therefore, the radius of the plastic zone *r_y_* rises.

For plane-stress conditions, the elastic strain energy release rate, *G*, was determined to be the following [28]:(3)G=KI2E
where the values of the elastic modulus, *E*, were found experimentally to be 26.94, 29.41, 32.30, 35.01, 40.35, and 46.00 GPa, respectively. 

Energy and power are used for non-periodic and periodic time signals. The square of the magnitude spectrum calculates a signal’s energy spectrum [29]. The highest EMR peak signal is supposed to be associated with the excited condition of the energised dislocations if they release EMR. The critical parameter would be the maximum EMR signal peak or *V_pmax_*. However, if the average cumulative impact of the dislocations is what causes the emission of EMR signals, then *V_rms_* must be a significant parameter. The highest *V_p_* part of the EMR signal was chosen to evaluate the average EMR energy release rate, G*_em_*. The maximum *V_p_* component of the signal has a time duration of *t*. The EMR voltage signal against the time graph was converted using MATLAB (r2016a) software into a *V_p_*^2^ versus *t* graph. The *G_em_* value was then determined [13] as
(4)$$G_{em}=\int \frac{{V_{p}}^{2}dt}{\Delta t}=\frac{Area\;below\;theV_{p-time,curve\;portion}^{2}}{∆t}$$

The maximum dominant EMR frequency, *f*, was calculated from the Fast Fourier Transform (FFT) function of Origin 2021, OriginPro by OriginLab Corporation software, which converted the time domain signals to frequency domain signals. The EMR frequency measures the dislocation strength and the media it moves through. Therefore, EMR frequency is a critical parameter to observe. 

## 3. Results and Discussion

### 3.1. Influence of 2a/w on EMR Signal Characteristics

The experimental findings, their interpretation, and the experimental conclusions are concisely and precisely described in this section. The plane-stress fracture toughness relates to the variables governing the metallurgical results and the geometrical parameters that influence the plane-stress fracture toughness [28]. As fracture toughness and EMR emissions are energy-triggered processes, the question of whether EMR emissions are linked to the specimen geometry arises.

Elastic–Plastic Fracture Mechanics (EPFM) considers the effects of plastic deformation near the crack tip and gives methodologies for evaluating crack growth and fracture toughness in materials displaying both elastic and plastic behaviour. Hence, we used notched samples so that crack initiation originates from the notches. Cracks profoundly impact the mechanical behaviour of materials, particularly in terms of fracture mechanics. Notches are geometric features intentionally incorporated into a specimen to create stress concentrations. For a sample of constant thickness and width, such as the one used in this study, increasing the notch length “2*a*” reduces the stress required to trigger additional crack extension.

EMR emission is related to plastic deformation, crack propagation, and fracture. Hence, it is vital to determine which signal in Nitinol specimens should be used for further analysis when the first EMR emission happens, and the nature of EMR emissions during progressive loading is a subject of investigation. Figure 3 shows the stress–strain plot depicting the stages of intermittent EMR emission for 2*a*/*w* = 0.33.

The EMR signal for various metallic materials [13,14,15,16,17,18,19] exhibits exponential decaying and damped oscillating characteristics. Figure 4a shows the first EMR emission signal near yielding for 2*a*/*w* = 0.33, indicating an oscillating damping nature. The EMR signal emitted from the Nitinol specimen at 2*a*/*w* = 0.33 at fracture is shown in Figure 4b. Here, the EMR signals showed an exponential decaying nature. The signals captured through the oscilloscope in real time give a better idea of the nature of the signal. Figure 4c,d provide a clear picture.

The oscillatory behaviour often occurs with increasing strain hardening. At higher strains, the signals become exponential. The strain value for the signal at 2*a*/*w* = 0.33 near yielding was 0.0189. For the fracture EMR signals at 2*a*/*w* = 0.33, the experimental tensile strain value was 0.060. The other EMR signals during tensile loading, i.e., signals 2 and 3, are also shown in Figure 4e,f, respectively.

During progressive strain hardening, the viscous damping coefficient of the alloy increases. The nature of the EMR released during the subsequently occurring phases of plastic deformation and propagation of cracks varies depending on the damping coefficient variation during loading, such as yielding, strain hardening, etc. The variable parameters under loading are the viscous damping coefficient and edge dislocation length due to possible dislocation when dislocation generation and multiplication occur. As a result, an investigation of EMR signal amplitude could help us comprehend the plastic deformation process with improved fidelity.

Intermittent EMR emissions showed a strong correlation with plastic deformation variables. As high-damping materials, shape memory alloys like Nitinol exhibit great potential as a valuable tool for both active and passive energy dissipation schemes in engineering.

The roots *p*_1n_ and *p*_2n_ of the governing equation for dislocation motion are given by the following [19]:(5)$$p_{1n,2n}=\frac{1}{2}\left[-c\pm \sqrt{c^{2}-4\left(d^{2}\right)}\right]$$

EMR amplitude/voltage can be expressed as follows [19]: (6)$$V_{0}\left(t\right)=\sqrt{N}\left[C_{1}\ddot{p}\left(t\right)+C_{2}\dot{p}\left(t\right)\right]$$
where *N* refers to total edge dislocations at the external surface of the specimen adjacent to the antenna. The EMR of the metal initiates from the area close to the surface. *C*_1_ and *C*_2_ are the antenna dimension parameters. *t* denotes time, *p*(*t*) is the electric line dipole moment, and the dots represent time derivatives. 

According to the physical model, when *c*^2^ > 4*d*^2^, the EMR voltage’s nature is predicted to be exponential, with real roots *p*_1n_ and *p*_2n_; when *c*^2^ < 4*d*^2^, the EMR voltage is predicted to be oscillatory, with complex roots. For the above cases,
*c* = *B*/*m*(7)
where *B* is the viscous coefficient of metal, *m* is the mass per unit length of edge dislocation *l*, and *d*^2^ = (2*n* + 1)^2^ π^2^D/*l*^2^, where *n* denotes the harmonics 0, 1, 2. D is the *T*_D_/*m*, where *T*_D_ is the line tension.

#### 3.1.1. Variation in EMR Parameters with Tensile Strain at Fracture Load

As previously indicated, the initial EMR signal (around the yield) provides the basis for the vital dislocation mechanism of the emission of EMR signals event. As deformation proceeds, new dislocations multiply. Also, the edge dislocations become highly energetic. Edge dislocations produce strain energy; some are lost through EMR signal emission. Hence, it is crucial to study the circumstances that lead to the highest energy emission from EMR signals throughout the tensile loading of the test specimens. Figure 5 shows the variations in *V_pmax_* and *G_em_* with tensile strain at the fracture load.

The EMR signal with the highest rate of electromagnetic energy release, *G_em_*, is the optimal EMR signal (linked with the maximum EMR amplitude, *V_pmax_*). The maximum value of EMR signal emission was always detected before the maximum load. The mechanical and EMR parameters observed during experimentation are listed in Table 3 for the EMR signals with varying 2*a*/*w* values.

#### 3.1.2. Correlation between Plastic Zone Radius and EMR Parameters

The variations in *V_pmax_*, *G_em_*, and *f* with the plastic zone radius *r_y_* for the maximum EMR emission signal for different values of 2*a*/*w* are shown in Figure 6 and Figure 7, respectively. With progressive stress, the plastic zone provides a platform for subsequent EMR emissions, accompanied by crack propagation and eventual fracture. 

There is a distinction between Figure 5 and Figure 6. The *r_y_* value was calculated from Equations (1) and (2) by the original value of 2*a*/*w* and the instantaneous value of tensile stress at maximum EMR emission (Table 3). The *V_p_* and *G_em_* values rose sharply for maximum EMR emission during strain hardening. During strain hardening, the total edge dislocations in EMR signals increase due to the operations of new Frank–Read sources. Figure 6 shows that not all edge dislocations are pinned up to a certain ry. Moreover, the pinning barriers (impurity clouds, dislocation networks, etc.) increase with further edge dislocations. This produces more edge dislocations to be pinned down, which raises the threshold for *V_p_* and *G_em_*. 

The mixed-frequency components of the EMR emissions produced by the plastic deformation of metallic alloys are already well established [6,8,16]. However, the kHz range corresponds to the highest energy burst frequency. The frequency is inverse to the EMR amplitude shown in the *f* versus *r_y_* plot (Figure 7).

#### 3.1.3. The Interrelation between the Energy Releases of the Mechanical and EMR Parameters

Figure 8 shows that *G_em_* correlates well with the *G* (coefficient of determination (R^2^) value of 0.9989). The “*G_em_*” value measures the energy dissipation through EMR emission during loading. *G* is directly proportional to the square of *K_I_*. Hence, measuring the *G_em_* value at fracture and its correlation with *G* helps evaluate the fracture toughness, a critical fracture mechanics parameter.

## 4. Microstructural Analysis

The microstructural analysis was performed based on the effect of notches on the fracture behaviour in Nitinol [30]. Figure 9a shows the Nitinol specimen before the test, and 9b shows the fractured surface.

The SEM images of the Nitinol specimen before fracture at higher magnification (×2500) were analysed through JEOL-JSM-6390. As shown in Figure 10, the porosity concentration was low, and the pore depth was a few micrometres; i.e., there was not much invasion. The porosity level was minimal, with shallow pores and no oxides. Oxides often have distinct surface textures compared to pure metals or other materials. They may appear granular and rougher in texture under SEM imaging. The surface roughness variations were minimal, thus indicating the absence of oxides.

Figure 11 shows the FE-SEM microstructure images of the Nitinol specimen at tensile fracture near the notch area for various 2*a*/*w* values at a higher magnification (2.5 K×), captured through the ZEISS 300, Carl Zeiss Microscope Ltd., Oberkochen, Germany, machine. The fractography analysis studied the crack initiation process at varying notch geometries, i.e., notch–width ratios. The fracture is characterised by void creation, expansion, and formation. The pronounced and dominating nature of the cleavage facets indicates brittle cleavage fracture. Analysing the fracture mechanisms is more challenging because the stress-induced phase transforms from austenite to deformed martensite during fracture. Dislocations influence the plastic deformation stages. 

The fractography in Figure 11 shows that the layer is disturbed, and there is flex-like breakage (broken layer). The behaviour described above is less present in Figure 11d, while the amount of breakage is lower in Figure 11f. Generally, breakage should originate from grain boundaries, but the complete layer/plane has come out; hence, the lattice planes have misaligned. Lattice planes represent geometric planes formed by groups of lattice points within the crystal lattice. The misalignment of lattice planes near the tensile fracture zone indicates crystal defects, i.e., grain boundaries, which suggest that the intermolecular energy has disturbed the crystal structure. This misalignment can affect the physical and mechanical properties of materials, as the arrangement of crystal planes influences properties like tensile strength. Therefore, the plane broke quickly into pieces. During tensile testing, the material reached a high temperature and vice versa, which did not allow the material to break in the form of flex in Figure 11f due to the sintering effect.

Also, due to a lattice structure mismatch of Ni (FCC structure) and Ti (closed pack hexagonal structure) and the addition of some other impurities during the fabrication of Nitinol, the crystal alignment was not supported during tensile fracture. The proper intermixing temperature during manufacturing may be one of the causes of this type of breakage behaviour. The constituent elements did not reach the melting temperature and, hence, disturbed the structure of the alloy element. This is the reason for this kind of flex breakage at the time of the tensile fracture. 

The fractography at tensile fracture for different notch–width ratios shows the specimens’ porosity variation, as seen in Figure 11. Figure 11a–c show more notable and substantial voids than those in Figure 11d,f, contributing to the weakening of the material. Porosity, the volume of intergranular pores, decreases as the material’s yield strength and ultimate tensile strength increase. Figure 11d (1294.16 MPa) (2*a*/*w* = 0.42) and Figure 11f (1349.25 MPa) (2*a*/*w* = 0.50) show higher tensile strength than Figure 10a–c. The cleavage facets alongside the surface are noticeable more in Figure 11a–c but cannot be differentiated in Figure 11d,f. This is due to increased tensile strength. The microstructure’s cleavage facet extension indicates material retention at the grain boundaries before breaking, which increases energy release through EMR emission. Investigations into the fracture behaviour of compact, thin Nitinol tension specimens have revealed that the cleavage fracture predominates as the primary fracture mechanism [30].

Cracks originate at the notch tip and subsequently expand and propagate. A plastic zone develops before the crack tip as the uniaxial stress increases. The cracked point creates a newer plastic zone ahead of each subsequent crack tip. In metals, dislocations create long-range electric fields in their vicinity. The accelerated motion of dislocations, as observed during the transformation process, is the probable cause of the emission of EMR [6]. When materials are subjected to complicated stress conditions, structures tend to fracture at stress-concentrated points such as cracks and notches. In semicircular notched samples, smaller notch sizes exhibited more brittle fractures [30]. 

Microvoids are common casting defects that affect metals’ tensile strength, quality, and integrity after casting. The nucleation and growth of microvoids affect the stress and strain fields near the crack tip. As shown in Figure 11, it is evident that where the voids/vacancies are in the majority, the tensile strength is weaker, and conversely, with fewer voids, the tensile strength is higher. The mechanical strength of Nitinol and the EMR energy release rate (*G_em_*) are directly correlated (Table 3). The *G_em_* value for the tensile fracture of Nitinol is lower at (28.5 V^2^ at 2*a*/*w* = 0.08) when the tensile strength of the alloy is lower at 1192.52 MPa. At a higher *G_em_* value (122.9 V^2^ at 2*a*/*w* = 0.50), the tensile strength of the alloy approaches its highest value (1349.25 MPa). This shows that the “*G_em_*” value is influenced directly by vacancies/voids, which affect tensile strength.

Metallic alloys emit intermittent EMR at the beginning of plastic deformation and during crack propagation. This is because, under external stress, accelerated electric line dipoles are produced by edge dislocations in the plastic zone ahead of a crack tip [31]. This study examined the correlation between the characteristics of EMR emission during tensile deformation and the fractography of Nitinol specimens (with different notch–width ratios) that depict failure and fracture mechanisms of materials. The deformation-induced EMR emissions in the experimental investigations were used as precursory signals to exhibit the investigation’s efficacy in deformation monitoring, failure forecasting, and comprehension of the complex fracture mechanism.

## 5. Conclusions

This investigation reports new insights into the influence of notch width on the EMR emission signals of Nitinol sheet specimens during tensile loading. Correlations between the mechanical and EMR emission parameters were discovered. The experimental findings led to the following broad conclusions: The EMR amplitude value rose linearly with increasing plastic zone radius, suggesting that pinning barriers become more prevalent once the plastic zone size reaches a certain threshold. New Frank–Read sources permit dislocation multiplication during strain hardening, rising with the plastic zone radius, increasing the maximum EMR amplitude and the average EMR energy release rate. Investigating the relation between EMR parameters and the plastic zone radius provides vital information for monitoring crack growth in materials and alloys at the microstructural level.The nature of the EMR emission signals near the yield and at fracture was observed to have oscillatory and exponential characteristics. With increased strain hardening, the viscous damping coefficient of the alloy rises, causing the signals to become exponential at a higher load value.The viscous damping coefficient and the edge dislocation length are variable parameters during loading due to probable dissociation during dislocation generation and multiplication. Hence, an analysis of the nature of EMR amplitude could facilitate comprehending the plastic deformation process at the atomic level with complementary molecular dynamics studies.A direct correlation was observed between the EMR energy release rate (*G_em_*) and the elastic strain energy release rate (*G*). By examining the EMR emitted upon fracture, the smooth relationship between *G_em_* and *G* was inferred as a novel method to evaluate the fracture toughness of metallic materials.

## Figures and Tables

**Figure 1 materials-17-01708-f001:**
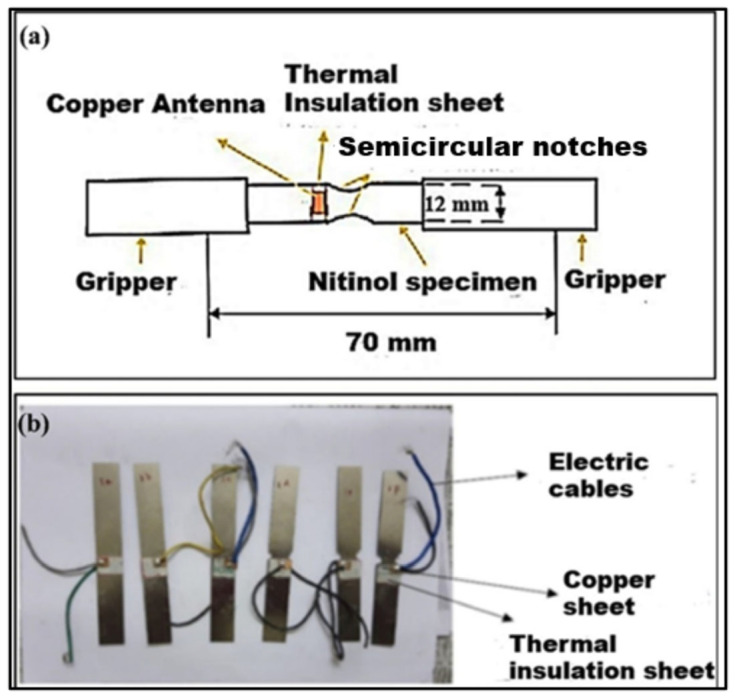
(**a**) A schematic of the specimen configuration (all dimensions are in mm) and (**b**) Nitinol test specimen used for experimentation.

**Figure 2 materials-17-01708-f002:**
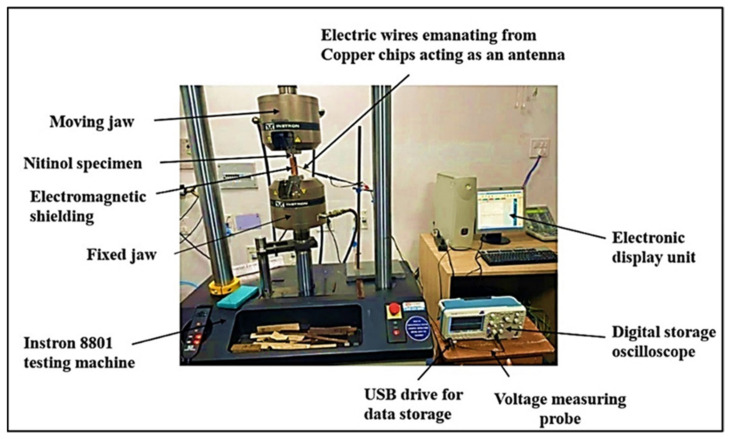
Experimental setup used to capture the EMR emission signals.

**Figure 3 materials-17-01708-f003:**
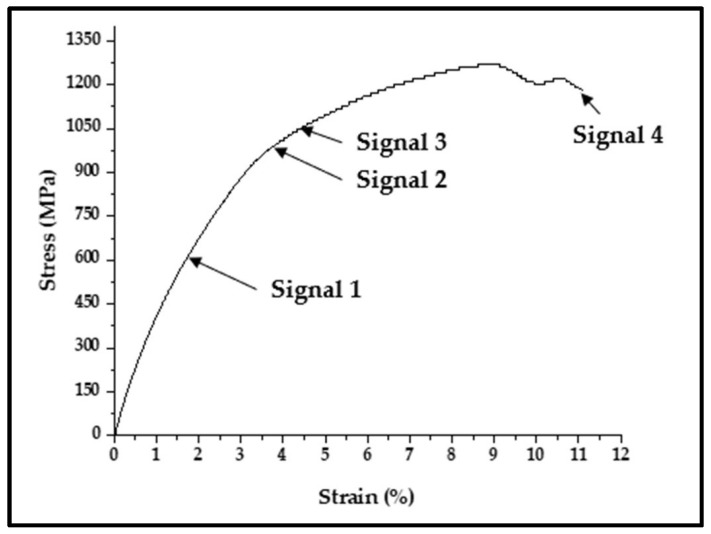
Stress–strain curve depicting the stages of intermittent EMR emission for 2*a*/*w* = 0.33.

**Figure 4 materials-17-01708-f004:**
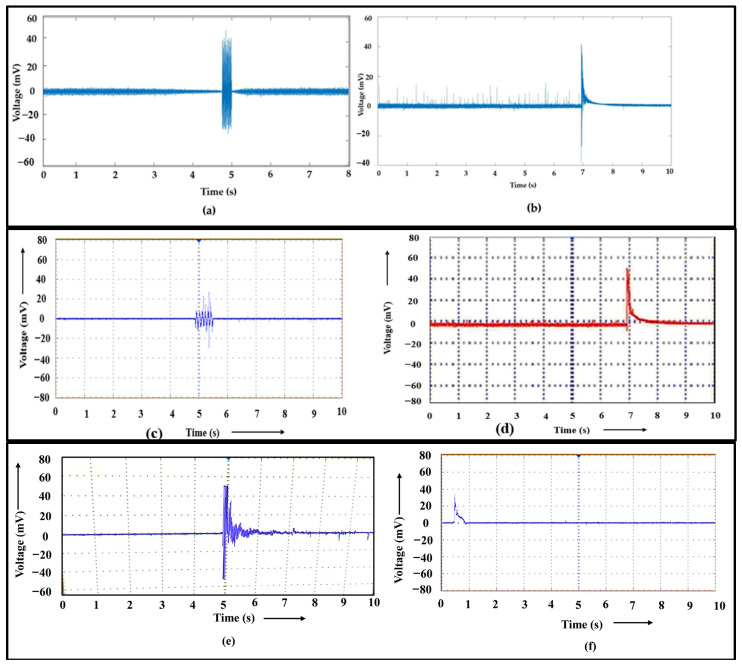
(**a**) EMR emission signal of the Nitinol specimen near yielding for 2*a*/*w* = 0.33. (**b**) EMR emission signal of Nitinol specimen at fracture for 2*a*/*w* = 0.33. (**c**) Real-time images of the EMR emission signal of the Nitinol specimen near yielding for 2*a*/*w* = 0.33. (**d**) EMR emission signal of Nitinol specimen at fracture for 2*a*/*w* = 0.33 captured through oscilloscope. (**e**) EMR emission signal 3 of the Nitinol specimen, and (**f**) EMR emission signal 4 of the Nitinol specimen during tension loading for 2*a*/*w* = 0.33.

**Figure 5 materials-17-01708-f005:**
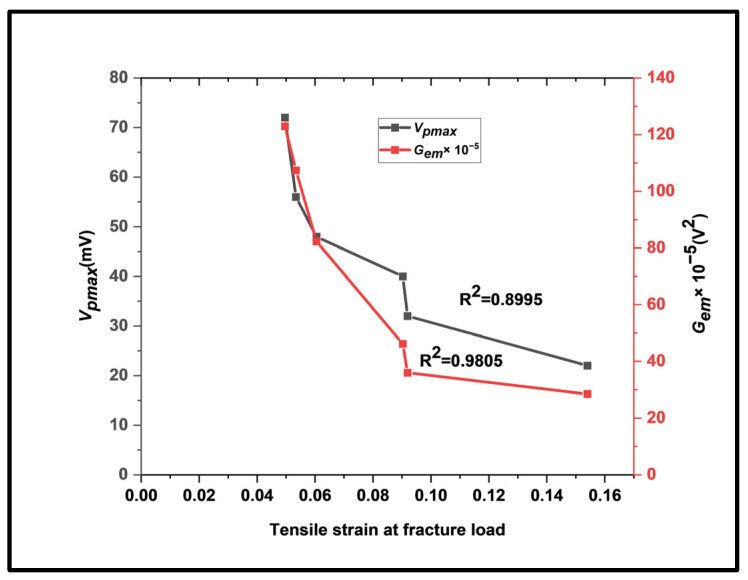
A plot showing the influence of *V_pmax_* and *G_em_* with tensile strain at fracture load.

**Figure 6 materials-17-01708-f006:**
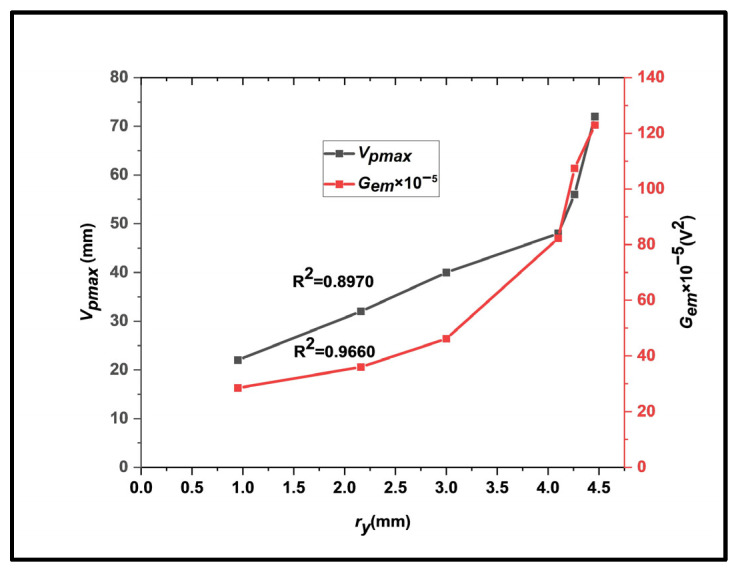
A plot showing the influence of *V_pmax_* and *G_em_* with *r_y_*.

**Figure 7 materials-17-01708-f007:**
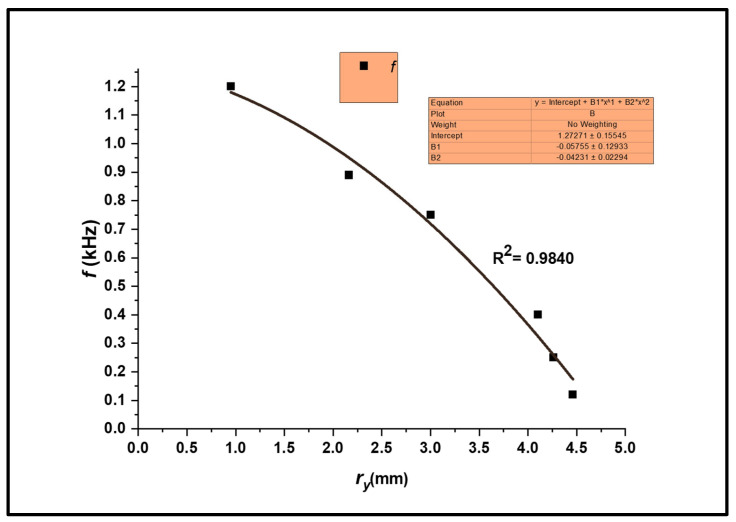
A plot showing the influence of *r_y_* on *f.*

**Figure 8 materials-17-01708-f008:**
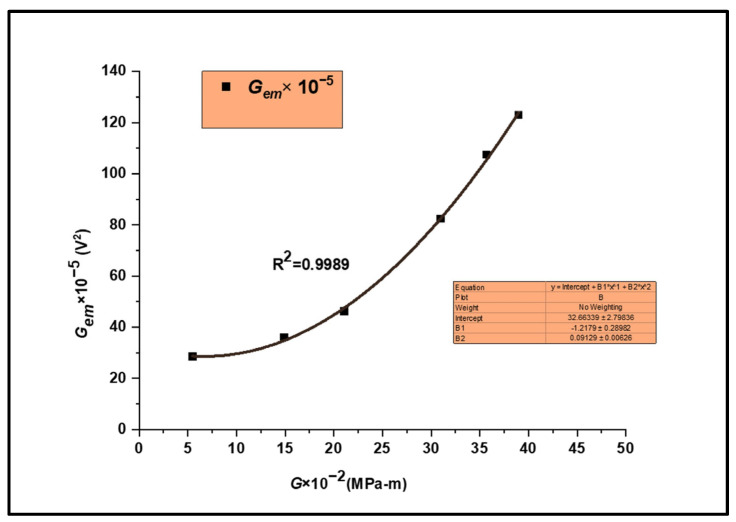
A plot showing the influence of *G* on *G_em_*.

**Figure 9 materials-17-01708-f009:**
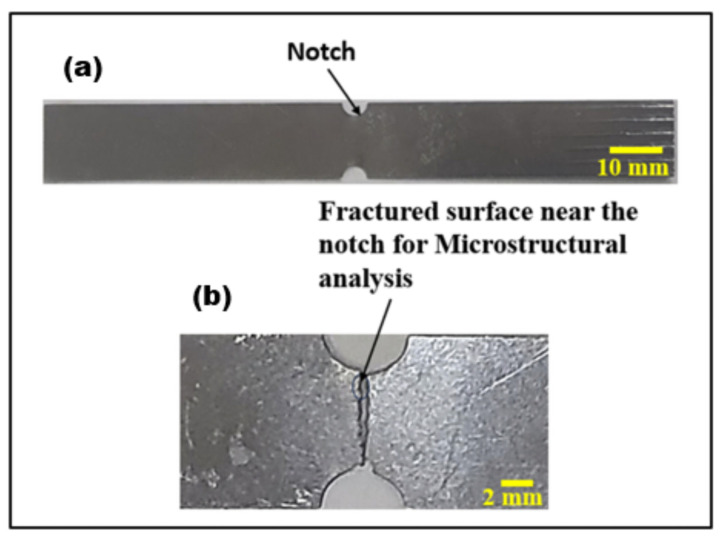
(**a**) The Nitinol specimen prior to the test and (**b**) the fractured surface.

**Figure 10 materials-17-01708-f010:**
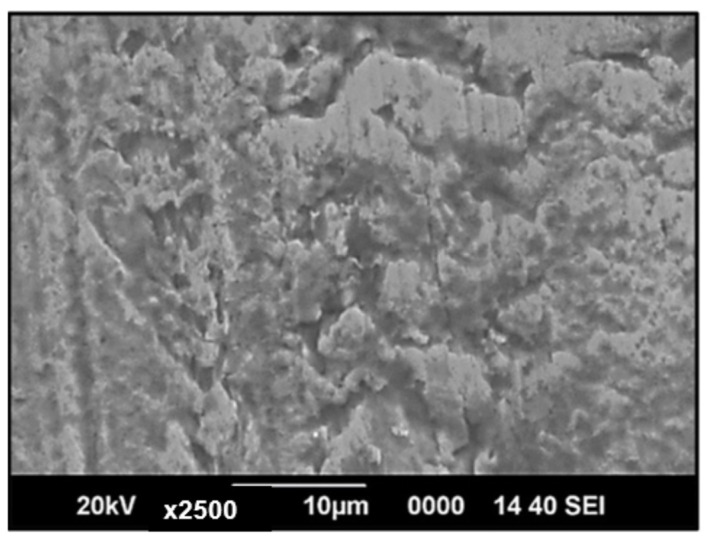
SEM micrographs of Nitinol before tensile fracture at higher magnification (2.50 K×).

**Figure 11 materials-17-01708-f011:**
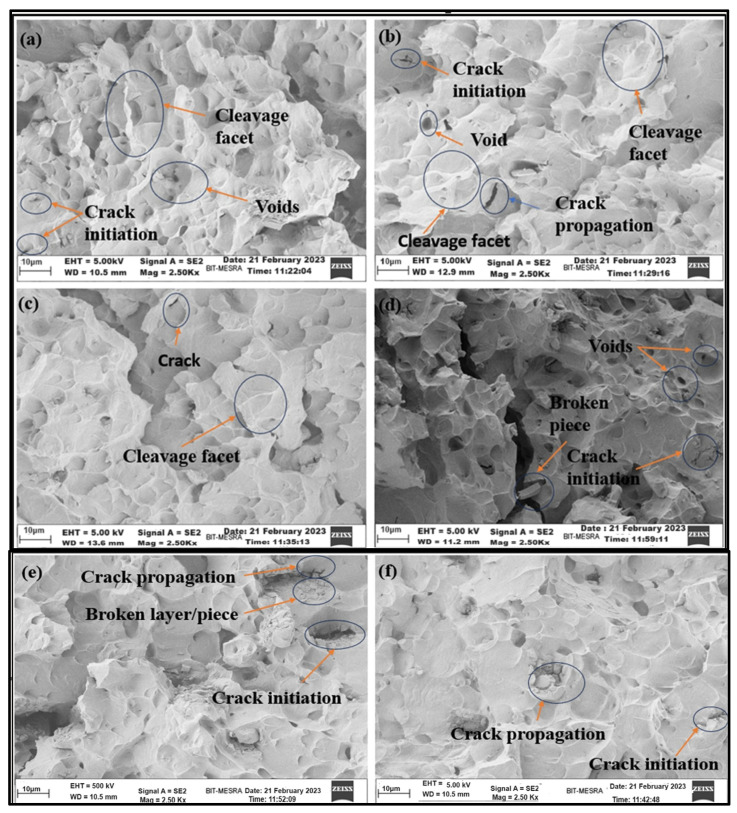
FESEM micrographs of Nitinol specimens during tensile fracture at higher magnification (2.50 K×) for 2*a*/*w* = (**a**) 0.08, (**b**) 0.16, (**c**) 0.25, (**d**) 0.33, (**e**) 0.42, and (**f**) 0.50.

**Table 1 materials-17-01708-t001:** Chemical composition of Nitinol (weight %).

Ni	Co	Cu	Cr	Fe	Nb	C	H	O	Ti
55.7	0.05	0.01	0.01	0.05	0.025	0.036	0.001	0.02	Balance

**Table 2 materials-17-01708-t002:** Input parameters for the experiments.

Notch Length (*a*) (mm)	Specimen Width (*w*) (mm)	Notch–Width Ratio (2*a*/*w*)	Specimen Thickness (mm)	Gauge Length (mm)	Loading Rate (mm/min)
0.5	12	0.08	0.5	70	5
1	12	0.16	0.5	70	5
1.5	12	0.25	0.5	70	5
2	12	0.33	0.5	70	5
2.5	12	0.42	0.5	70	5
3	12	0.50	0.5	70	5

**Table 3 materials-17-01708-t003:** Mechanical and EMR parameters for different notch–width ratios of Nitinol.

Notch-Tip Radius (*a*)	Notch–Width Ratio (2*a*/*w*)	Fracture Load (N)	Tensile Stress Fracture Load (MPa)	Tensile Strain at Fracture Load	Critical Stress Intensity Factor (*K_C_*) (MPa−m^1/2^)	*r_y_* (mm)	*G* × 10^−2^ (MPa−m)	*V_p_*_max_ (mV)	*f* (kHz)	*G_em_* × 10^−5^ (V^2^)
0.5	0.08	6558.95	1192.54	0.1540	38.69	0.95	5.50	22	1.20	28.5
1	0.16	5761.11	1152.23	0.0919	66.10	2.16	14.9	32	0.89	36
1.5	0.25	5651.26	1255.84	0.0903	82.56	3.00	21.1	40	0.75	46.2
2	0.33	5544.71	1386.18	0.0605	112.40	4.10	31.0	48	0.40	82.3
2.5	0.42	4529.56	1294.16	0.0534	120.10	4.26	35.7	56	0.25	107.4
3	0.50	4047.76	1349.25	0.0496	133.99	4.46	39.0	72	0.12	122.9

## Data Availability

Data can be obtained by writing to the authors.

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
