# Peer review of "Influence of an Engineered Notch on the Electromagnetic Radiation Performance of NiTi Shape Memory Alloy"

_materials, 2024, doi:10.3390/ma17071708_

Round 1

Reviewer 1 Report

Comments and Suggestions for Authors

This paper discussed the impact of the notch size on the EMR during alloy specimen tensile test, which can be used as an evaluation factor on specimen’s deformation. From my opinion this paper can be published on Materials after following issues are addressed.

1.      The authors mentioned multiple samples were tested to get consistent EMR signals. Could the authors provide discussion on result repeatability and distribution?

2.      Could the authors provide the stress-strain curves (Figure 3) and EMR emission signals (Figure 4) of other samples? Such plots can be placed into supplementary if necessary.

3.      In Figure 10, the authors claimed “the porosity concentration was low, and the pore depth was a few micrometers”. A visual comparison between SEM images of quantitative comparison is needed to support the discussion.

Reviewer 2 Report

Comments and Suggestions for Authors

The manuscript reports on the synthesis of nitinol (NiTi) with double semicircle edge notches with different notch lengths and on the study of their stress, strain and EMR behavior. The objective is to use EMR as a non-destructive tool for analyzing crack initiation and propagation in metallic materials, understand the tensile fracture behaviour of Nitinol and to shed light on the relationship between crack initiation, fracture behaviour, and EMR emission. The manuscript has original results, but needs revisions. I have the following comments:

- In the abstract is written “The EMR parameters remained constant initially but rose sharply with the plastic zone radius with progressive loading.”. What are these parameters ?

- On page 2 it is written “It changes into martensite at lower temperatures, a more complicated “monoclinic” (FCC crystal structure).”. A monoclinic structure cannot be FCC (face centered cubic). It is either monoclinic or cubic. It cannot be both at the same time.

- Figure 3 indicates several signals. Where were they presented in the paper ?

- On page 6 it is written “Figure 4 (a) shows the first EMR emission signal near yield for 2a/w = 0.33, indicating an oscillating damping nature.”. How is the oscillatory behavior seen in the figure ? What happens at ~5s in fig 4a and ~7s in figure 4b ?

- The legends in figures 5, 6, 7 and 8 are not informative. These legends refer to variables that are not in the axis and is confusing (e.g., in figure 5 the legend refers to Vpmax and Gem, but the axis are named with longer names not showing the Vpmax and Gem). They should be more clear. Also, the legend of figure 7 is “A plot showing the influence of f on ry”, but it should be “A plot showing the influence of ry on f” (it is the other way). The same happen in the legend of figure 8.

- Figure 7 shows a fitting curve. What was the fitting function ? The same for figure 8.

- The images of figure 9 should have scale bars.

- On page 10 it is written “The presence of oxides was missing.”. How was that determined ? From chemical analysis ? Where is it ?

- On page 11 it is written “hence, the lattice indices have misaligned, indicating that the intermolecular energy has disturbed the crystal structure”. What does the sentence “hence, the lattice indices have misaligned” mean ?

- On page 12 it is written “As shown in Figure 11, the tensile strength decreases where vacancies are predominant and conversely”. What does conversely mean in this context ?

Reviewer 3 Report

Comments and Suggestions for Authors

The research explores the impact of the notch width on Electromagnetic Radiation (EMR) performance in Nitinol Shape Memory Alloy during tensile tests, revealing correlations between EMR parameters and plastic zone radius. The topic is important for monitoring of crack initiation and fracture in NiTi SMAs. However there are critical question regarding Fracture theory used:
1- the employed theory is for cracked samples. However, you has only notched specimen. Moreover, I am unsure if the linear elastic fracture theory can be applied to such an elastic-plastic problem

2- Why for each specimen a specific yield stress has been reported? Sy shall be a material property and independent to the geometry.

3- Normally by reducing the a/w ratio (from deep to shallow crack) the geometrical constraints at the crack tip decrease resulting in higher plasticity at the crack tip (i.e., higher ry value). However, the reported ry values in Table 3 are not following this concept.

Comments on the Quality of English Language

I noticed minor typos and grammar errors.

Round 2

Reviewer 2 Report

Comments and Suggestions for Authors

The manuscript reports on the synthesis of nitinol (NiTi) with double semicircle edge notches with different notch lengths and on the study of their stress, strain and EMR behavior. The current manuscript is a revised version that has answered most of the questions and only needs minor revisions. I have the following comments regarding the raised questions:

Comment: C6. Figure 7 shows a fitting curve. What was the fitting function? The same for figure 8.

Authors answer: R6. The fitting curves for Figures 7 and 8 have been included in the manuscript on Page 11.

Comment: The equations in the graphics of figures 7 and 8 do not seem complete. It seems something is missing in them.

Comment: C9. On page 11 it is written “hence, the lattice indices have misaligned, indicating that the intermolecular energy has disturbed the crystal structure”. What does the sentence “hence, the lattice indices have misaligned” mean ?

Authors answer: R9. Misalignment of lattice indices near the tensile fracture zone indicates crystal defects, so that the intermolecular energy has disturbed the crystal structure. This misalignment can affect the physical and mechanical properties of materials, as the arrangement of crystal planes influences properties like tensile strength.

The misalignment of lattices has been explained in Section 4, Page 13, First Paragraph

Comments: When the authors refer to “Misalignment of lattice indices” it seems the authors are referring to misalignment of lattice planes as they refer to them in the end of the sentence. It would be better to speak about lattice planes, as it is more standard.

Author Response

Pl see attached 

Reviewer 3 Report

Comments and Suggestions for Authors

Please add KC (critical KI value at the fracture load) to table 3. Please also report the notch-tip radios which you assumed as a crack.

Comments on the Quality of English Language

Minor typos and grammar errors.

Author Response

pl see attached.
